# Precision Nutrition for Alzheimer’s Prevention in *ApoE4* Carriers

**DOI:** 10.3390/nu13041362

**Published:** 2021-04-19

**Authors:** Nicholas G. Norwitz, Nabeel Saif, Ingrid Estrada Ariza, Richard S. Isaacson

**Affiliations:** 1Harvard Medical School, Boston, MA 02115, USA; 2Department of Neurology, Weill Cornell Medicine and NewYork-Presbyterian, New York, NY 10065, USA; nas2782@med.cornell.edu (N.S.); ingridestrada1113@hotmail.com (I.E.A.); rii9004@med.cornell.edu (R.S.I.)

**Keywords:** Alzheimer’s disease, ApoE4, precision nutrition, astrocytes, microglia, blood–brain barrier, inflammation, insulin resistance

## Abstract

The *ApoE4* allele is the most well-studied genetic risk factor for Alzheimer’s disease, a condition that is increasing in prevalence and remains without a cure. Precision nutrition targeting metabolic pathways altered by *ApoE4* provides a tool for the potential prevention of disease. However, no long-term human studies have been conducted to determine effective nutritional protocols for the prevention of Alzheimer’s disease in *ApoE4* carriers. This may be because relatively little is yet known about the precise mechanisms by which the genetic variant confers an increased risk of dementia. Fortunately, recent research is beginning to shine a spotlight on these mechanisms. These new data open up the opportunity for speculation as to how carriers might ameliorate risk through lifestyle and nutrition. Herein, we review recent discoveries about how *ApoE4* differentially impacts microglia and inflammatory pathways, astrocytes and lipid metabolism, pericytes and blood–brain barrier integrity, and insulin resistance and glucose metabolism. We use these data as a basis to speculate a precision nutrition approach for *ApoE4* carriers, including a low-glycemic index diet with a ketogenic option, specific Mediterranean-style food choices, and a panel of seven nutritional supplements. Where possible, we integrate basic scientific mechanisms with human observational studies to create a more complete and convincing rationale for this precision nutrition approach. Until recent research discoveries can be translated into long-term human studies, a mechanism-informed practical clinical approach may be useful for clinicians and patients with *ApoE4* to adopt a lifestyle and nutrition plan geared towards Alzheimer’s risk reduction.

## 1. Introduction

Alzheimer’s disease (AD) is ascending the list of the top ten causes of death in the United States and remains the only one without a cure. Even with respect to symptom management, 99.6% of drug candidates fail [1]. At this pace, the number of Americans living with AD is projected to triple to 13.8 million by mid-century [2], imposing an unsustainable economic toll and an unimaginable human burden. 

Pathological changes due to AD can begin to develop in the brain decades before any clinical symptoms manifest [3]. As emerging evidence continues to identify modifiable risk factors for AD, it is increasingly understood that sporadic AD, which accounts for 95% of cases, may be considered to be a lifestyle disease. In other words, lifestyle factors that contribute to the fundamental metabolic pathologies of AD can increase risk of experiencing cognitive decline and dementia. 

The group at greatest genetic risk are those carrying the *ApoE4* allele, which codes for a mutated form of apolipoprotein E (ApoE). ApoE protein plays an essential role in the transport and metabolism of lipids throughout the body, including the brain where it is synthesized primarily by astrocytes, but also be other glial cells. The *ApoE4* allele is the leading genetic risk factor for late-onset AD. Although only approximately 20% of the general population carry *ApoE4*, carriers account for an estimated 40–65% of AD cases [4]. Depending on whether a carrier has one or two copies, *ApoE4* increases AD risk in the general population by three to 12-fold [5]. 

While *ApoE4* is associated with the two neuropathological hallmarks of AD, amyloid plaques and tau tangles, relatively little is known about precisely how *ApoE4* contributes to disease risk. Nevertheless, gene–environment interaction are thought to play a mediating role. For example, *ApoE4* is common in persons of West African ancestry but does not meaningfully contribute to AD risk in those living in West Africa [6,7]. The Indianapolis–Ibadan dementia project, a longitudinal prospective study including 4606 individuals of African descent living in either the United States or Nigeria, found that the incidence of AD was twice as high in Africans living in the United States [8]. The Mediterranean diet and lifestyle are also associated with cognitive longevity. Strikingly, Southern Italians who carry *ApoE4* and live in Italy exhibit a normal rate of living to be in the oldest 1% of the population, above 95 years for men and 99 years for women; odds ratio 1.21, *p* = 0.38. However, Southern Italians who carry *ApoE4* but who live in the United States exhibit a highly reduced chance of living into late old age; odds ratio 0.29, *p* < 0.01 [9]. This is especially meaningful given that the median survival duration after AD diagnosis is 3.1–5.9 years [10]. Comparisons such as these suggest that environment and lifestyle mediate the effect of *ApoE4* on the development of AD. 

The beneficial impact of a healthy lifestyle on disease course has been confirmed in several randomized controlled trials, such as the FINGER, MAPT, ENLIGHTEN, and SPRINT-MIND studies. However, these studies will not be a focus of this manuscript for four reasons. (i) First, each of these studies employed a multidomain approach (recommendations for diet, exercise, cognitive engagement, etc.). While this is critical to maximally reduce AD risk in a clinical setting, it obscures the potential effects of nutrition specifically. (ii) Second, these studies implemented interventions in higher-risk populations and in older-aged cohorts. Median age at baseline for all studies was at least 65 years old and included individuals who were overweight, hypertensive, diabetic, sedentary, or smokers [11,12,13,14]. For a disease process that begins at middle age or before, lifestyle interventions and metabolic health may need to be initiated decades earlier. (iii) Third, only the FINGER study performed an *ApoE4* subgroup analysis [15]. However, the multidimensional nature of the study interventions minimizes its relevance to this manuscript. Additionally, the median age of carriers at baseline was 69.2 years, the majority were overweight or obese and hypertensive, and a minority were diabetic and smokers. (iv) Fourth, the nutritional recommendations used in the FINGER and other trials were based on general health guidelines rather than on the principles of precision medicine, which the National Institutes of Health defines as “an emerging approach for disease treatment and prevention that takes into account individual variability in genes, environment, and lifestyle for each person” [16]. 

Herein, we discuss the future of precision nutrition for AD prevention in *ApoE4* carriers. Our intention is to consider the problem from a broader angle that could inform future clinical trials for the prevention of AD in persons at high genetic risk. Given the noted limitations of the clinical data, we build a case from biological principles by first considering novel literature on how *ApoE4* alters microglia, astrocyte, pericyte and vascular function, as well as insulin resistance. We chose to focus on these glial cells and insulin resistance because this is where the tide of recent literature on *ApoE4* has turned, and sensibly so. Microglia are the resident immune cells in the brain, and it is increasingly clear that AD is an inflammatory disease [17,18]. Astrocytes are the main producers of ApoE4 in the brain as well as a primary center of cerebral lipid metabolism [19,20]. Pericytes influence blood–brain barrier (BBB) function [21] and it now appears that *ApoE4* influences AD progression by directly altering vascular function [22]. Finally, insulin resistance seems particularly pertinent in *ApoE4* carriers [23,24] and provides a clear target for nutritional intervention. Throughout, we most heavily emphasize data from *ApoE4*-target replacement mice as controlled human trials investigating mechanisms are difficult or impossible to conduct and the pathways dissected are usually highly conserved.

After reviewing important recent findings in the literature, we suggest precision nutrition interventions that may help to reduce risk for AD in *ApoE4* carriers. The efficacy of the combined interventions is not, at present, quantifiable. Clinical outcome data are not available and, though blood biomarkers should be followed over time [25], these may not be specific to carriers. Thus, this manuscript is limited to be a biologically plausible discussion and set of suggestions. Nevertheless, it is our hope that this manuscript is useful for clinicians and their patients, as well as an informative starting point for the development of future clinical trials.

## 2. Microglia and Inflammation

Microglia are the brain’s resident immune cells and principally responsible for the neuroinflammation that is a necessary component of AD. The emergence of a pro-inflammatory, disease-associated microglia (DAM) phenotype is thought to be an early step in the evolving amyloid cascade hypothesis in which amyloid beta (Aβ) pathology induces tau pathology. 

Several lines of evidence suggest DAM could be a bridge between early Aβ oligomers and tau tangles. Prior studies have shown that reactive microglia drive tau pathology in animal models [26] and that reactive microglia are associated with tauopathy in humans [27]. In mice, intracerebroventricular injections of small Aβ oligomers are sufficient to induce microglial activation [28], while inhibition of the NLRP3 inflammasome (an essential component of the DAM phenotype [29]) protects against amyloid-induced tauopathy and cognitive decline [30]. These data are consistent with the notion that DAM link Aβ to tau. There is also murine evidence that microglial NLRP3 assembly itself can serve as a nucleating core for early amyloid plaques and that blocking microglial NLRP3 can prevent the spreading of Aβ [31]. Thus, as NLRP3 appears to be essential in DAM activation and Aβ and tau pathology, it may serve as a promising target for preventative interventions. 

Additionally, in animal models, microglial activation is an early and necessary step of AD pathology observed during the pre-plaque stage [32]. Correspondingly, in humans, microglial activation can be observed via PET scan in individuals with MCI, sometimes despite a lack of amyloid tracer uptake [33]. Conversely, no evidence of DAM was observed in asymptomatic persons with increased amyloid levels [34]. These data collectively suggest microglial-mediated inflammation is an early feature of AD pathogenesis and that it is a necessary step for the progression of disease to the symptomatic state. Simply put, a brain without inflammation may well be a brain without AD. (For a more comprehensive recent review of microglia in AD, see Leng and Edison [35]). 

The questions most relevant to this manuscript are, does *ApoE4* impact microglia function and, if so, what are the consequences? *In vitro* work using dissociated neuron-microglia cultures and hippocampal slices from transgenic *ApoE4*-replacement mice has demonstrated that microglia-mediated neuronal damage is worse in the presence of *ApoE4* [36]. In mice, an ApoE protein-dependent pathway mediates the transition of microglia from a homeostatic to a DAM phenotype [37,38] and, correspondingly, the *ApoE4* allele is associated with more reactive inflammatory microglia and worse tau pathology [39]. Furthermore, CRISPR/Cas9-mediated conversion of human iPSC-derived microglia from *ApoE3* to isogenic *ApoE4* is associated with a change in the expression of 1460 genes, in association with impaired ability to clear Aβ_42_ and an inflammatory transcriptomic signature [40]. These experiments strongly suggest that *ApoE4* alters microglia function to promote the DAM phenotype and advance disease.

Recently, Friedberg and colleagues [17] placed the above data in human context by performing post-mortem analyses on individuals from the Framingham Heart Study cohort, 35 of whom were *ApoE4* carriers. Interestingly, microglial density was associated with increased tau pathology in *ApoE4* carriers only, not in non-carriers. Furthermore, the protective effect of certain anti-inflammatory cytokines present in non-carriers was erased in *ApoE4* carriers in conjunction with a six-fold stronger association between frontal cortex tau pathology and dementia in carriers. Taken in the context of the iPSC and murine data, these new findings advance a model in which *ApoE4* pathologically alters microglial behavior and the inflammatory response to contribute to the development of AD. It also provides a target for precision nutritional intervention.

## 3. Astrocytes and Lipid Metabolism

Astrocytes are the most numerous glial cells and provide structural and metabolic support to neurons. Recently, Habib and colleagues [41] defined a group of astrocytes characterized by a unique transcriptomic profile. They termed these cells disease-associated astrocytes (DAA) to parallel the inflammatory DAM phenotype described in the previous section. DAA are observed in AD, both in mice and in humans, and their abundance increases with disease progression. Interestingly, in this study, *ApoE* was one of only eighteen signature dysregulated genes shared between the DAM and DAA phenotypes [41]. This begs the question, could *ApoE4* exacerbate DAA pathology? Transgenic mouse data demonstrated that *ApoE4* primes the brain to react with a strong inflammatory signal in response to an inflammatory insult. Challenge with lipopolysaccharide endotoxin resulted in an NFκB-dependent upregulation in inflammatory genes [42]. This is notable in the context of DAA because obesogenic, inflammatory diets can induce NFκB and downstream markers of reactive astrocytes in model systems [41,43,44]. Thus, it is conceivable that *ApoE4* could create a diathesis for diet-induced DAA formation, a dysfunctional astrocyte phenotype largely specific to AD [41]. It is, however, worth remarking that a similar dysfunctional response was recently observed in *ApoE4* astrocytes infected by SARS-CoV-2 virus [45]. 

*ApoE4* can also cause endosomal dysfunction in astrocytes, resulting in a decrease in surface membrane levels of critical receptors, such as LRP1, a protein responsible for Aβ clearance. (LRP1 is also helps to inhibit the CypA-NFκB-MMP9 pathway, discussed below). Prasad and colleagues [46] documented, in mice, that *ApoE4* leads to an increase in endosome acidity in a histone-deacetylase (HDAC)-dependent manner. As a result, there was a decrease in LRP1 recycling to the surface membrane and decrease in Aβ clearance. This dysfunction could be rescued by HDAC inhibitors, including butyrate, to normalize endosomal acidity, recover LRP1 recycling and surface levels, and restore Aβ-clearance capacity to that of *ApoE3* astrocytes [46]. Importantly, the HDAC pathway implicated is highly evolutionarily conserved and can be manipulated by nutrient status [47], as will be discussed later in this manuscript. 

In addition to an increased susceptibility to the DAA phenotype and endosomal dysfunction, lipid metabolism is altered in astrocytes. Astrocytes are a center of lipid metabolism and cholesterol synthesis in the brain and its main producer of the lipoprotein ApoE4, coded by the *ApoE4* allele [48]. Below, we give four examples of ApoE4-mediated impairments in cerebral lipid metabolism and transport. 

First, early reports suggest that *ApoE4* astrocytes have impaired ability to synthesize and secrete cholesterol, which is essential in neuronal functioning [49,50]. The brain may only be 2% of the body’s mass, but it contains 25% of its cholesterol [51]. Furthermore, RNA-Seq performed on astrocytes from aging mice suggests that aging also decreases the ability of astrocytes to synthesize cholesterol, including a reduction in the rate limiting enzyme of cholesterol synthesis, HMG-CoA reductase [52]. Thus, *ApoE4* plus aging could equal a cerebral cholesterol deficit. Interestingly, this same mouse study also found an age-dependent increase in cholesterol transport genes, including *ApoE*, which could be the aging brain’s attempt at compensation. 

Second, part of the ApoE4 protein’s impaired lipid transport capacity may have to do with its poor lipidation status relative to that of ApoE3, as demonstrated nicely by Rawat and colleagues using target replacement mice [53]. The protein, ATP-binding cassette A1 (ABCA1), controls ApoE lipidation by transporting intracellular cholesterol and phospholipids to ApoE to form ApoE-HDLs in astrocytes. This process requires ABCA1 recycling between endosomes and the surface membrane, similar to LRP1 endosomal recycling as mentioned above. However, ApoE4 protein aggregates in endosomes along with ABCA1. As a result, there is less surface ABCA1, less ApoE lipidation, and a decreased lipid transport capacity in the brain. ABCA1 activation can rescue membrane surface level and cholesterol efflux function, and ABCA1 activators are being explored as a potential treatment for AD. Unfortunately, Rawat and colleagues did not investigate the role that endosome acidity plays in this process, nor whether HDAC inhibition could normalize surface ABCA1 levels and ApoE lipidation, as is the case with LRP1 [46]. In our opinions, it is a probable proposition that should be investigated further. 

Third, Qi and colleagues [20] recently found that *ApoE4* disrupts fatty acid metabolism coupling between neurons and astrocytes in an *in vitro* culture system, and this could contribute to an energy deficit and increase in AD risk. They showed that *ApoE4* astrocytes had an impaired ability to clear neuronal lipid droplets, in part, because ApoE4 protein impairs trafficking of lipid droplets [20] from neurons to astrocytes. This is consequential for two reasons. First, neurons are not efficient at β-oxidation, so *de novo*-synthesized long-chain fatty acids accumulating in neurons could contribute to lipotoxicity. Second, it creates a seemingly hopeless scenario with respect to energy metabolism. Not only can lipotoxicity impair mitochondrial function, but a decrease in lipid supply to astrocytes and decrease in astrocyte lipid metabolism could force a shift towards glucose metabolism. However, the group also showed that *ApoE4* can contribute to inhibition of pyruvate dehydrogenase (PDH), relative to *ApoE3*, thus creating a blockade in glucose metabolism (see the insulin resistance section for more details on cerebral glucose metabolism in *ApoE4* carriers). 

Thus, as the *ApoE4* brain ages and is unable to compensate energetically, it would appear that “metabolic crisis” is inevitable. However, this need not be the case as neuronal lipophagy can also clear lipid droplets, protecting neurons from lipotoxicity [20], and alternate sources of acetyl-CoA, including the ketone body β-hydroxybutyrate (βhB), can circumvent a PDH blockade and supply fuel substrate to neurons. This will be discussed more below in the precision nutrition section. 

Fourth, and finally, Li and colleagues [54] also published data showing *ApoE4* disrupts coupling of cholesterol metabolism between neurons and astrocyte in target replacement mice and that this has epigenetic and behavioral consequences. The data support a model in which ApoE particles secreted from astrocytes not only confer cholesterol to neurons, but also microRNA (miRNA). These miRNA function to downregulate cholesterol synthesis to preserve neuronal energy, thus shifting the burden of cholesterol synthesis from neurons to astrocytes. Furthermore, as neuronal acetyl-CoA is not unnecessarily shunted to *de novo* cholesterol synthesis, acetyl-CoA can accumulate, leading to an increase in histone acetylation and expression of immediate early genes important in memory consolidation. ApoE4 protein is worse at delivering miRNA, as compared to ApoE3 protein, resulting in impaired downregulation of cholesterol synthesis enzymes in neurons, lower relative acetyl-CoA levels, decreased histone acetylation, and decreased expression of memory genes. 

In summary of its effects on lipid metabolism, (i) *ApoE4* astrocytes have impaired ability to synthesize and secrete cholesterol [49,50], possibly contributing to an age-dependent cerebral cholesterol deficit. (ii) *ApoE4* decreases surface ABCA1 levels, resulting in impaired ApoE lipidation and lipid trafficking [53]. (iii) *ApoE4* decreases lipid droplet trafficking from neurons to astrocytes, possibly resulting in neuronal lipotoxicity and an energy deficit in astrocytes [20]. (iv) ApoE4 particles delivers less cholesterol and miRNA to neurons, forcing neurons to expend energy on cholesterol synthesis, taxing the acetyl-CoA pool, decreasing histone acetylation, and decreasing the expression of memory genes [54].

## 4. Pericytes and Blood–Brain Barrier

Pericytes are glial cells associated with the BBB, and pericyte dysfunction culminating in BBB breakdown can be a step in the pathogenesis of AD. Recently, Montagne and colleagues [21] published an important human study showing that *ApoE4* contributes to AD in this way. They demonstrated that *ApoE4* carriers are characterized by breakdown of the BBB prior to cognitive impairment and that high baseline markers of pericyte dysfunction in the cerebrospinal fluid (CSF) predicts future cognitive decline specifically in *ApoE4* carriers, and not in non-carriers.

Delving more deeply into the data, they showed that BBB breakdown begins in patients in the hippocampus and parahippocampal gyrus, regions important for memory. This precedes brain atrophy and can occur independent of systemic vascular risk and independent of amyloid and tau accumulation. Furthermore, elevated levels of pericyte injury marker, platelet-derived growth factor β (PGDFβ) in the CSF predicted accelerated cognitive decline on global mental status exams specifically in *ApoE4* carriers, a finding that was likewise independent of amyloid and tau levels. These data strongly suggest that pericyte dysfunction and BBB breakdown are independent risk factors for AD specifically in human *ApoE4* carriers [21]. However, is the dysfunction targetable?

Montagne et al. also identified the cyclophilin A-matrix metalloproteinase-9 (CypA-MMP9) pathway as a likely key contributor to BBB breakdown and cognitive decline in human *ApoE4* carriers. The CypA-MMP9 pathway is a pro-inflammatory cascade that can be initiated by pericytes and terminates in the expression of the MMP9 gelatinase enzyme that degrades the basement membrane and tight junctions. In human iPSC-derived pericytes, the Montagne group found substantially higher levels of CypA and MMP9 in *ApoE4* cells [21]. Complementary animal data suggest that the observed correlations are causal and that ApoE4-mediated increase in CypA-MMP9 contributes to BBB dysfunction and cognitive decline AD. Using transgenic mice, Bell and colleagues [55] showed that ApoE3 protein, but not ApoE4 protein, can bind to the LRP1 receptor to inhibit CypA expression and reduce downstream NFκB and MMP9 activity, protecting BBB integrity. Activation of the CypA-NFκB-MMP9 axis was six-fold higher in the *ApoE4* mice and preceded neuronal damage. Critically, pharmacological inhibition any step in the pathway—CypA, NFκB, or MMP9—was sufficient to prevent BBB dysfunction and the leakage of neurotoxic systemic factors into the brain [55]. 

While the CypA-NFκB-MMP9 will be the primary focus of the nutritional strategies discussed below, (as it may be the most novel, *ApoE4*-specific, and targetable pathway in pericytes at the present time), it is worth remarking that another manuscript was published the same month as the writing of this manuscript demonstrating that *ApoE4* can alter cognitive function by decreasing cerebral blood flow. Yamazaki and colleagues [22] observed that conditional expression of *ApoE4* in vascular mural cells in mice was sufficient to decrease cerebral blood flow, decrease long-term potentiation and learning, and induce adverse behavior changes [22]. Thus, an effective nutritional strategy for preventing AD in *ApoE4* carriers should focus on blocking components of the CypA-NFκB-MMP9 axis and optimizing cerebral flood flow. 

## 5. Insulin Resistance and Glucose Metabolism

Insulin resistance is a central feature of AD. Even preclinically, patients with AD show widespread impairment in cerebral glucose metabolism [56]. Cerebral hyperinsulinemia can further contribute to the development and propagation of amyloid and tau pathology. For example, insulin directly competes with insulin-degrading enzyme for Aβ degradation [57,58,59]. Furthermore, insulin resistance results in activation of GSK3β (a principle kinase involved in tau hyperphosphorylation, and which is also known as tau kinase I [60,61,62]), thereby contributing to the generation of more amyloid and tau pathology in a positive feedback cycle [63]. (For details on GSK3β-amyloid-tau positive feedback, please see our 2019 publication on a multi-loop model of Alzheimer’s disease [63]).

Interestingly, insulin resistance and *ApoE4* appear to have a synergistic effect. The relative risk of development of AD with type II diabetes alone is 1.8, whereas the additional presence of an *ApoE4* allele increases relative risk to 5.5 [24]. More impressively, PET scan studies reveal that cognitively normal middle-aged *ApoE4* carriers have impaired cerebral glucose metabolism, as compared to non-carriers, and in a dose-dependent manner [64,65]. These latter data suggest that *ApoE4* could impair insulin signaling to contribute to AD. Animal model data concur. 

Zhao and colleagues [23] performed a landmark study using *ApoE4* and *ApoE3* transgenic mice that demonstrated ApoE4 protein can impair insulin receptor signaling, compete with insulin for receptor binding, and trap insulin receptors in endosomes. They observed that, in aged *ApoE4* mice there was decreased inhibition of the insulin pathway target, GSK3/tau kinase I, in the cortex and hippocampus. ApoE4 bound the insulin receptor with higher affinity than ApoE3, competitively reduced insulin binding by twice as much as ApoE3, and decreased surface insulin receptor levels in correspondence with their entrapment in endosomes (again, similar to LRP1 and ABCA1). Importantly, the pathological phenotype just described was largely age dependent and accelerated by an obesogenic Western diet [23]. Thus, these data provide mechanistic insight that is not fatalistic, but rather informative in directing nutritional therapy, as will be discussed in the next section.

## 6. Precision Nutrition Considerations

In this section, we consider which dietary patterns and nutritional supplements may be particularly beneficial for *ApoE4* carriers. As there are currently no long-term prospective or interventional studies that investigate this topic, these considerations are based on the data presented above. As the mechanisms we discussed are drawn mostly from mouse models, they may be limited in their translatability to humans. This is an inherent, but important, limitation of our mechanism-first approach. Where appropriate, recommendations are put into context of correlations drawn from population-based studies to create the most informed precision nutrition approach given the current state of the data. 

### 6.1. Low-Glycemic Index and Low-Carbohydrate Diets

Nutrition for *ApoE4* carriers should focus on preventing insulin resistance. True insulin resistance is usually marked by hyperinsulinemia and an increase in fasting insulin (with the exception of β-cell failure in type II diabetes) and coincides with sub-optimal metabolic heath. Insulin resistance is a pathology inherent to the metabolic syndrome and, as over one-third of Americans have metabolic syndrome [66] and 88% exhibit at least one of its markers [67], optimizing insulin sensitivity should be a dietary priority. This can be achieved through low-glycemic index [68] or low-carbohydrate diets [69,70,71]. Interestingly, it has even been proposed that high-carbohydrate diets contribute to AD on the basis of meta-analyses that reveal lower frequency of *ApoE4* in populations with long-standing historic exposure to agriculture, such as the Turks or Mayans (*ApoE4* allele frequency of 0.079–0.089), as compared to hunter-gatherers such as the Papuans or Inuits (0.214–0.368) [72]. Additionally, as mentioned in the introduction, Nigerians consume one-quarter of the amount of sugar that Americans consume and exhibit low rates of AD despite a high prevalence of *ApoE4* [6,8,72].

Optimizing insulin sensitivity by eating a low-glycemic index diet throughout the life course may help to prevent cerebral hyperinsulinemia and competition of Aβ degradation by insulin, prevent GSK3β hyperactivity and tau hyperphosphorylation [63], and avoid exacerbating the potential effects of ApoE4 protein on the insulin cascade. Recall that Zhao and colleagues [23] found that the negative effects of *ApoE4* on insulin resistance in their mice were accelerated by an obesogenic Western diet. Furthermore, avoiding or reversing insulin resistance by eating a low-carbohydrate diet can help to improve hypertension and presumably optimize cerebral blood flow [73], as was shown to be especially important in *ApoE4* carriers in the recent Yamazaki study [22], discussed above.

Low-glycemic index and low-carbohydrate diets may also help in AD prevention by preventing the glycation of ApoE. Glycation of ApoE containing particles impairs their uptake [74] and, thus, likely reduces their ability to traffic cholesterol and lipids in the brain. Furthermore, ApoE4 protein has 3-fold the binding affinity for advanced glycation end products (AGEs) [75], as compared to that of ApoE3, and AGEs are observed at three-fold higher levels in the amyloid plaques of individuals with AD as compared to those from healthy controls [76]. Thus, it may well be that glycation in the brain contributes to AD, not only by contributing to Aβ and tau pathologies, but also by altering other aspects of metabolism. Consequently, while it remains speculation, it is plausible that the enhanced glycation potential of *ApoE4* contributes to impaired brain lipid metabolism, Aβ monomer oligomerization, plaque deposition, and disease progression, and that by reducing glycation through eating a low-glycemic or low-carbohydrate diet, carriers can reduce their risk. 

### 6.2. Ketogenic Diets and β-Hydroxybutyrate

On the spectrum of low-glycemic index and low-carbohydrate diets, a more advanced option for interested patients may be a ketogenic diet. Ketogenic diets are a form of therapeutic carbohydrate restriction in which carbohydrate intake is reduced to a level (usually below ~25 g of net carbohydrates per day) such that the liver converts body and/or dietary fat in the ketone body, βhB. Diet-induced nutritional ketosis (βhB > 0.5–5 mM) is distinct from diabetic ketoacidosis and ketogenic diets are complementary with many other dietary strategies, including vegetarian, carnivore, or Mediterranean diets. 

Although this particular manuscript aims to focus on disease prevention, it is worth noting that dietary interventions that increase βhB have shown to be effective at improving symptoms in individuals with AD, in part, by supplying an alternative source of fuel, as opposed to glucose, to the brain [77,78,79]. For example, a recent randomized controlled crossover trial comparing a 12-week ketogenic diet to a 12-week low-fat diet found that the former caused significant improvements in daily functioning and overall quality of life as compared to the latter [80]. We choose to comment briefly on these symptom data both to further the rationale for providing a ketogenic option to at-risk patients and to acknowledge that there is a lack of consensus over whether *ApoE4* carriers constitute a less responsive group to ketogenic therapy. For example, one study found that administration of the ketogenic substance, caprylic acid, to individuals with diagnosed AD led to improvements only in non-carriers [81]. There are at least two important caveats to this finding. First, the degree of ketosis induced was light and temporary. Even directly post-supplementation, βhB levels only reached 0.4 mM, which is short of nutritional ketosis and may not have been sufficient to compensate for the presumably larger impairment in glucose metabolism in *ApoE4* carriers. By contrast, a ketogenic diet has been shown to improve cognition in *ApoE4* context [79]. Additionally, in support of the notion that βhB utilization is normal in carriers, Wu et al. found that, while glucose metabolism is impaired by *ApoE4* in cell and mouse models, as compared to *ApoE3* controls, ketone metabolism is increased relative to *ApoE3* [82]. This finding is consistent with the possibility that *ApoE4* genotype shifts metabolic dependency from glucose to ketones and, therefore, a ketogenic diet could be protective against *ApoE4*-associated metabolic deficits and reduce AD risk. 

Ketogenic diets could help reduce AD risk via manifold biological plausible mechanisms. βhB can decrease inflammation, oxidative stress, mitochondrial dysfunction, and other basic metabolic pathologies that contribute to metabolic diseases, including AD [83]. As these metabolic pathologies are engaged in a complex positive feedback network, it is important that any preventative intervention has the potential to modulate metabolism at each node of the network [63,83]. This may be particularly important in vulnerable *ApoE4* carriers. For example, it has been proposed that oxidative stress induces expression of *ApoE4* in neurons as a repair response and that ApoE4 protein is more vulnerable than ApoE3 protein to proteolytic cleavage, generating a C-terminal ApoE4 fragment that may damage mitochondria [84,85,86]. Pre-existing mitochondrial dysfunction, inflammation, and insulin resistance would sensitize or exacerbate the positive feedback network. However, what if an individual reduced the oxidative stress, inflammation, and insulin resistance through lifestyle? This is the fundamental basis of *ApoE4* precision nutrition—relieve the metabolic burden to reduce risk.

βhB may also provide a solution to the “metabolic crisis” discussed with respect to the Qi et al. study [20] in the above section on astrocytes. A ketogenic diet could have the following three effects. First, it could protect against lipotoxicity in neurons by enhancing neuronal lipophagy [87], and this could be further enhanced by an intermittent fasting protocol [88]. (As an aside, the higher dietary fat intake of a ketogenic diet would not itself contribute to lipotoxicity. In a human randomized controlled three-arm crossover trial, Hyde et al. showed that, even when total and saturated dietary fat intakes are increased by 2–3-fold, serum triglycerides and serum saturated fat still decreased relative to isocaloric higher-carbohydrate, lower-fat diets [70]. Perhaps more to the point, long-chain fatty acid transport into the brain is tightly regulated.) Second, as a low-carbohydrate diet, a ketogenic diet could potentially improve ApoE-mediated lipid transport by decreasing lipoprotein particle glycation. Third, βhB is an alternate source of acetyl-CoA [89] that does not require PDH or β-oxidation to sustain the brain’s metabolic needs. Additionally, in hippocampal cells, βhB concentrations achievable through diet protect against amyloid toxicity [90].

Further on the topic of astrocyte–neuron lipid coupling, βhB could compensate for the effects of ApoE4 particles’ impaired ability to deliver regulatory miRNA from astrocytes to neurons, which include lower acetyl-CoA levels, less histone acetylation, and decreased expression of memory genes [54]. βhB could increase miRNA delivery by increasing LRP1 levels [91], which itself promotes ApoE-mediated delivery of miRNA [54]. βhB increases acetyl-CoA levels and inhibits HDACs to increase histone acetylation and potentially increase memory gene expression. In brief, mechanisms exist whereby βhB could compensate for the defects in astrocyte–neuron lipid coupling described recently [20,54].

Turning to inflammation, βhB inhibits the formation of the NLRP3 inflammasome [92,93] that plays a critical role in the DAM phenotype and progression of amyloid and tau pathology in AD [30,31], as discussed above in the section on microglia. Indeed, Shippy and colleagues recently found that βhB inhibited the NLRP3 inflammasome, prevented microgliosis, and prevented plaque formation in a mouse model of AD [92]. 

βhB may further block the other inflammatory cascade discussed above, the CypA-NFκB-MMP9 pathway, that is responsible for BBB breakdown and predictive of cognitive decline specifically in *ApoE4* carriers. The pathway is inhibited by LRP1, a receptor that is also responsible for Aβ clearance and that is decreased by ApoE4 in a HDAC-dependent manner, as discussed in conjunction with Prasad study above [46]. βhB is an HDAC inhibitor [94,95], providing a mechanistic basis for βhB-mediated suppression of CypA-NFκB-MMP9 cascade via an increase in the pathway inhibitor, LRP1, which would also support Aβ clearance. Indeed, βhB was recently shown to increase LRP1 levels in a human BBB model [91]. Furthermore, βhB inhibits the NFκB component of the pathway by binding GPR109A receptor [96,97], which the Montagne group found was sufficient to protect the BBB in *ApoE4* mice [21]. Additionally, a ketogenic diet in mice was recently shown to inhibit HDAC, NFκB, and MMP9 expression [98]. 

In sum, interventions to promote ketosis have potential to target two key inflammatory processes—NLRP3 inflammasome activation promoting DAM, and the CypA-NFκB-MMP9 promoting BBB breakdown—that are otherwise elevated in carriers of *ApoE4*. 

Finally, while no long-term prospective trials for the potential of dietary interventions to prevent AD in *ApoE4* carriers exists, a recent human study observed that ketogenic diets and βhB improved a newly established marker of brain aging, brain network stability, whereas carbohydrate-rich diets impaired brain network stability [99]. Thus, mechanisms and emerging data suggest that adopting a lifestyle on the spectrum of low-glycemic index, low-carbohydrate, and ketogenic diets might help to protect against insulin resistance, inhibit pathological inflammatory processes (NLRP3 and CypA-NFκB-MMP9), and be an overall effective neuroprotective strategy in carriers of *ApoE4*. 

### 6.3. Mediterranean Dietary Components

The Mediterranean diet is associated low rates of AD and improved disease progression [100,101], with one study suggesting that high adherence can reduce risk of cognitive impairment by 33% [102]. Impressively, as mentioned in the introduction, carriers of *ApoE4* who lived in Italy, and presumably ate a traditional true Mediterranean diet for most of their life, were as likely as non-carriers to live to be above 95 for men and 99 for women. By contrast, genetically similar Italian *ApoE4* carriers living in the United States had a reduced chance of living into late old age (odds ratio 0.29, *p* < 0.01) [9]. These association data are interesting but made more compelling when put into the context of mechanisms. 

#### 6.3.1. Extra Virgin Olive Oil

Extra virgin olive oil contains neuroprotective phenolic compounds, including oleocanthal and hydroxytyrosol, that have been shown to have anti-amyloid and anti-tau properties [103,104,105,106]. Such compounds may also impact pathways specifically impaired in *ApoE4* carriers. For example, administration of oleocanthal to mice was found to increase levels of the CypA-NFκB-MMP9 inhibitor protein, LRP1, in brain capillaries by 27% in association with reduced AD pathology [107]. Additionally, both oleocanthal and hydroxytyrosol were shown, in human vascular cells, to decrease NFκB and MMP9 activity by 50–80% [108]. Furthermore, in vitro incubation of macrophages with olive oil polyphenols increases ABCA1 levels in a dose-dependent manner and, more importantly, twelve weeks of extra virgin olive oil consumption in humans was shown to increase ABCA1 expression by 16%, in conjunction with an increase in cholesterol efflux capacity [109]. It is possible that a similar effect in the brain could help improve astrocyte lipid metabolism. In brief, the epidemiology on the Mediterranean lifestyle and the presence of multiple mechanisms by which olive oil and its constituents could help to prevent AD make a reasonable case that it could play a protective role in *ApoE4* carriers’ diets.

#### 6.3.2. Certain Vegetables

Compounds found in certain low-carbohydrate vegetables may also synergize with those in extra virgin olive oil. For example, quercetin, an antioxidant found in highest concentrations in capers and red onions, likewise decreases NFκB and MMP9 activity [108]. Given that a typical Western diet contains on the order of 10 mg of quercetin, one would only need to consume two tablespoons of capers to quintuple median intake [110,111], although supplementation may be warranted to achieve a therapeutic dose (see below). Sulphoraphane too, found in cruciferous vegetables such as broccoli, broccoli sprouts, cauliflower, Brussels sprouts, and arugula, can decrease NFκB activity and MMP9 expression by as much as seven-fold [112,113]. 

#### 6.3.3. Fatty Fish

Fatty fish are a final important part of Mediterranean diets, with wild salmon, mackerel, and sardines being perhaps the most healthful and available. These foods are a rich source of the two long-chain omega-3 fatty acids, eicosapentaenoic acid (EPA) and docosahexaenoic acid (DHA). DHA in particular is essential for brain function. Brain DHA levels are lower in individuals with AD as compared to controls [114,115] and greater consumption of fish and DHA is associated with lower rates of AD [116,117]. These associations are backed by animal models in which DHA can improve Aβ and tau pathologies [118,119]. DHA can inhibit NLRP3 [120], correspondingly tune microglia function [121], and DHA can inhibit NFκB-MMP9 activity [122]. Thus, it is possible that DHA targets pathways adversely affected in *ApoE4* carriers and may prevent DAM-mediated inflammation and pericyte dysfunction leading to BBB breakdown. 

DHA dose and timing also matter. *ApoE4* carriers may need higher DHA doses because they β-oxidize DHA at greater rates than non-carriers [123]. In addition, carriers are more likely to exhibit BBB dysfunction [21] that can impair DHA delivery to the brain [124]. These two factors may explain why some (but not all) studies have found a negative association between fatty fish intake and the development of AD in the general population, while no association was observed in *ApoE4* carriers [125]. Importantly, a PET imaging study performed on young (mean age 35) healthy individuals observed a 16% increase in DHA incorporation into whole brain gray matter, with a 34% increase in the entorhinal cortex, in *ApoE4* carriers as compared to non-carriers. This was hypothesized to be a compensatory mechanism for increased DHA utilization and increased metabolic demands also observed in carriers [126]. Thus, those wishing to adopt a conservative approach may choose to consume more than the standard recommendation of two servings of fatty fish per week and may opt to additionally supplement (see below). 

#### 6.3.4. Limit Alcohol

While light alcohol consumption has been associated with a decreased risk of AD in general [127], this relationship does not appear to hold in *ApoE4* carriers. Consumption of any amount of alcohol may increase the risk of AD for *ApoE4* carriers [128,129,130]. In one study, both light and moderate alcohol consumption were associated with improvement in learning and memory for non-carriers, but with a decline in learning and memory for carriers [129]. Other studies found that *ApoE4* carriers who consumed alcohol one or more times per month had a higher risk of AD than those who never consumed alcohol [128] and the risk of AD for carriers increased with increasing amounts of alcohol consumption [130] These data suggest that alcohol consumption should be limited, especially in *ApoE4* carriers. 

### 6.4. Supplementation

#### 6.4.1. DHA

As mentioned above, DHA supplementation may be desirable in *ApoE4* carriers to ensure an adequate dose. Krill oil and triglyceride-DHA may be a potential consideration although further studies are warranted. Krill oil is rich in phosphatidylcholine-conjugated DHA and may have favored access to the brain via the MSFD2A transporter [131]. Triglyceride-DHA is the most bioavailable and most effective at increasing serum DHA concentrations [132]. Given the likely increased DHA needs of *ApoE4* carriers, 2 g/day may be considered a minimum dose.

#### 6.4.2. Quercetin

As noted above, the phenolic compound quercetin can inhibit NFκB and MMP9 activity [108]. It can also significantly decrease NLRP3 inflammasome formation [133,134,135,136]. Thus, quercetin from food and as a supplement may to protect BBB integrity and prevent the activation of DAM in *ApoE4* carriers. On the basis of the doses used in the referenced animal model studies, 25–100 mg/kg/day, it may be clinically judicious to supplement in addition to eating a healthy diet. As rodents have moderately faster metabolisms than humans, an over-the-counter dose of 1–2 g/day may be sufficient, but further studies are warranted to come to more definitive conclusions. 

#### 6.4.3. Resveratrol

Resveratrol, popularly presented as ‘the red wine polyphenol’ and sirtuin activator, has been shown in human AD patients, at 2 g/day, to reduce CSF MMP9 levels by approximately one-third as compared to placebo, in association with an attenuation of cognitive decline [137]. However, the sample size in this study was small (n = 15) and more research is needed to determine recommended dose. Resveratrol can also inhibit the NLRP3 [138,139] and diminish DAM and DAA inflammatory phenotypes [139,140,141]. 

In addition, resveratrol can activate the master regulator of mitochondria biogenesis, PGC1α in a sirtuin-dependent manner [142]. This is particularly notable in the *ApoE4* context because *ApoE4*, as compared to *ApoE3*, exhibits reduced glucose metabolic capacity; however, this can be rescued by increasing PGC1α expression in model systems [82]. Thus, high-dose resveratrol, similar to βhB but by a complementary and probably less potent mechanism, could help *ApoE4* neurons avoid “metabolic crisis” in which glucose and lipid metabolism are both impaired. βhB and intermittent fasting could help protect against lipotoxic damage to mitochondria by promoting lipophagy and also provide an alternative acetyl-CoA precursor for the generation of energy via the Krebs cycle. At the same time, resveratrol-induced PGC1α expression could facilitate further mitochondrial biogenesis and potentially improve deficient glucose metabolism. 

Since the average wine contains less than 2 mg/L of resveratrol, one would need to drink approximately 8 standard barrels of wine to achieve the 2 g/day dose [143]. While further studies are needed to delineate evidence-based dosing suggestions, the above human AD did use 2 g/day [137], and doses as high as 5 g/day have been used safely [144]. 

#### 6.4.4. Vitamins D_3_ and K_2_

A combination of vitamin D_3_ and K_2_ is a consideration for people living at higher latitudes. Vitamin D_3_ is an important steroid hormone evolutionarily produced upon full body exposure to direct sunlight. However, given modern human lifestyles, few people endogenously produce adequate levels of vitamin D_3_, and as many as 77% of Americans may have insufficient levels [145]. As for K_2_, the menaquinone 7 (MK7) form is most active and difficult to obtain through diet, although found in Japanese Natto (fermented soybean) and some fermented cheeses [146].

D_3_ and K_2_ target pathways beneficial for bone health and cardiovascular risk [146], D_3_ and K analogs can inhibit NLRP3 inflammasome formation [147,148] and D_3_ can inhibit MMP9 expression [149,150,151]. Furthermore, vitamin D can increase ABCA1 levels when patients are vitamin D insufficient, helping to promote cholesterol efflux capacity and, possibly, astrocyte lipid metabolism [152,153]. 

In typical patients, a minimum serum 25-hydroxyvitamin D_3_ level of 30 nmol/L would be considered a sufficient target. However, studies have suggested that a higher target range (50–70 nmol/L) may produce greater cognitive benefits, and this is particularly the case for *ApoE4* homozygotes in whom higher levels of serum 25-hydroxyvitamin D_3_ improved memory function [154,155]. Most patients can tolerate 5000 IU vitamin D_3_ and a range from 45 to 180 µg/day K_2_-MK7, although further studies are necessary to more clearly guide clinical practice. 

#### 6.4.5. B Vitamins

Elevated homocysteine (Hcy) is a risk factor for AD [156], positively associated with *ApoE4* status [157], and it is generally accepted that lower Hcy levels are desirable. Interestingly, Hcy can decrease *ApoE* gene expression in an NFκB-dependent manner [158]. Since ApoE is required for cholesterol transport within the brain, and *ApoE4* astrocytes already exhibit impaired transport ability [49,50], elevated Hcy could exacerbate a metabolic deficiency. Supplementation with B-vitamins can improve Hcy levels [159]. 

In the VITACOG randomized controlled trial, including 26% *ApoE4* carriers, B-vitamin complex supplementation slowed brain atrophy by 70% in those with high baseline Hcy, but only in the tertile with the highest serum omega-3 levels. Correspondingly, high baseline omega-3 levels were associated with decreased brain atrophy, but only in the B-vitamin supplementation group and not with the placebo [160]. These data suggest that the combination of omega-3 and sufficient B-vitamins might act synergistically to slow cognitive decline. Data such as these suggest could also explain why omega-3 supplementation does not show benefits in all studies. 

B-vitamins may also synergize with fatty fish intake and DHA supplementation to prevent brain atrophy. Recall that iPSC-derived *ApoE4* astrocytes exhibit a dysregulation in key genes involved in phospholipid metabolism [40]. Enzymes involved in phospholipid metabolism are vital for the transport and incorporation of DHA into the brain [161,162] and these enzymes can be inhibited by a metabolic precursor to Hcy [163]. Furthermore, in rats, B-vitamins can increase DHA concentrations in the blood [164]. Collectively, these data suggest B-vitamins might be essential for proper Hcy control and DHA utilization. 

Furthermore, vitamin B_2_ (riboflavin), B_3_ (niacin), B_6_, and B_12_ have each been shown to inhibit either or both NLRP3 and NFκB, with niacin potentially acting by binding the same GPR109A receptor as βhB, mentioned above [165,166,167,168]. 

For these reasons, *ApoE4* carriers may consider supplementing with a full-spectrum B-vitamin complex including, at minimum, 0.8 mg folate as methyltetrahydrofolate (5-MTHF), 0.5 mg B_12_ as methylcobalamin, and 20 mg of B_6_ as pyridoxal 5′ phosphate (PLP). These doses were derived from the VITACOG randomized trial [160]. Some practitioners may further choose to suggest the specific forms of these B-vitamins, mentioned above, that are more bioactive and particularly important for the half of the population that carry the common *MTHFR* C677T or A1298C alleles, which impair B-vitamin metabolism and increase Hcy levels [169]. 

#### 6.4.6. Lithium

As discussed above, individuals with *ApoE4* may be at increased risk for insulin resistance, and downstream GSK3β/tau kinase I hyperactivity. Overactive GSK3β can contribute to Aβ and tau pathologies as well as further insulin resistance via positive feedback pathways [63]. Furthermore, animal data suggest that GSK3β inhibition by lithium can increase Aβ clearance from the brain, in part, by increasing levels of the CypA-NFκB-MMP9 pathway regulator, LRP1, at the BBB [170]. Lithium is a recognized GSK3β inhibitor [171] with an interesting epidemiological relationship to AD. Large scale studies on Scandinavian and American populations have found that higher levels of trace lithium in drinking water are associated with lower incidences of AD and AD-related mortality [172,173]. A small interventional trial involving patients with mild AD showed that microdosing of lithium successfully prevented cognitive decline over fifteen months as compared to a control group [174]. By contrast, most human trials administering lithium to patients with manifest AD do not show improvement in symptoms. However, symptom management is not synonymous with prevention, and it remains an open question whether long-term exposure to lithium could help to prevent disease [175]. Epidemiology and mechanisms appear to suggest this is possible. Hopefully, future studies will answer this question while also stratifying groups by genotype to determine whether the intervention is more or less effective in *ApoE4* carriers. 

## 7. Ethical Considerations

This manuscript is not meant to advocate broad genetic testing for *ApoE4*. Given the lack of robust clinical evidence for precision nutrition in the context of AD risk reduction, such a practice would be premature. Furthermore, it is important to acknowledge that *ApoE4* is a risk allele likely influenced by gene–gene and gene–environment interactions that might also modify the potential efficacy of the dietary suggestions presented in this text. That said, there may be utility in providing biologically plausible suggestions, as we have done in this manuscript, if for no other reason than direct to consumer genetic testing is becoming more widely available and many patients choose to know their genotype. Potential downsides of direct-to-consumer testing include the potential for misinterpretation of results by patients, as well as a potential negative psychological reaction to the information [176]. 

However, the Risk Evaluation and Education for Alzheimer’s Disease (REVEAL) trial was the first randomized controlled to evaluate the impact that disclosure of *ApoE4* carrier status had on behavioral change in cognitively normal individuals and the researchers found that individuals who learned they were carriers reported more behavioral changes related to diet, exercise, medications, and vitamins compared to those who learned they were non-carriers [177]. This study demonstrates that the act of disclosing *ApoE4* carrier status can positively affect behavior change, which is critical for AD risk reduction success. That said, patients do not require knowledge of their *ApoE* status to implement the dietary or supplement options provided in Table 1 and Table 2. It is also likely that several of the suggestions, such as consuming a low-glycemic index diet and fatty fish, are likely to benefit the cognitive longevity of most persons. Thus, a conservative patient–clinician team could implement dietary options without genetic testing.

In sum, while *ApoE4* carrier status is a sensitive topic that certainly requires a collaborative discussion between the patient and treating clinician, in cases deemed clinically appropriate by both parties (and where the patient does not already have the information through direct-to-consumer testing), disclosing this information may be a beneficial way to encourage behavioral change. The specifics of these changes in diet and lifestyle should be determined only following a thoughtful discussion between the patient and the clinician that considers the individual’s sociocultural and economic circumstances, as well as any other practical limitations in implementing lifestyle change and the personal preferences.

## 8. Conclusions

This manuscript proposes elements of a precision nutrition approach to reduce risk of AD in individuals carrying the *ApoE4* allele (Figure 1, Table 1 and Table 2). While there are few human studies showing that targeted nutritional interventions can prevent AD, it may serve as a useful steppingstone to future clinical research that can in turn help to further guide care. We based our recommendations on a careful look at the recent data on *ApoE4* biology, which emphasizes the impact of *ApoE4* on glial cell function, energy metabolism, and insulin resistance. 

Finally, while this manuscript focused on nutrition, it is essential to understand that well-informed nutritional choices are only one prong of a complete preventative lifestyle. Adequate sleep, proper exercise, stress reduction, and social connectedness are important in reducing the risk of developing AD [25,178]. *ApoE4* is a risk gene and it is environment that converts genetic risk into disease. The still open question for the future is what lifestyle is required to defuse *ApoE4*? 

## Figures and Tables

**Figure 1 nutrients-13-01362-f001:**
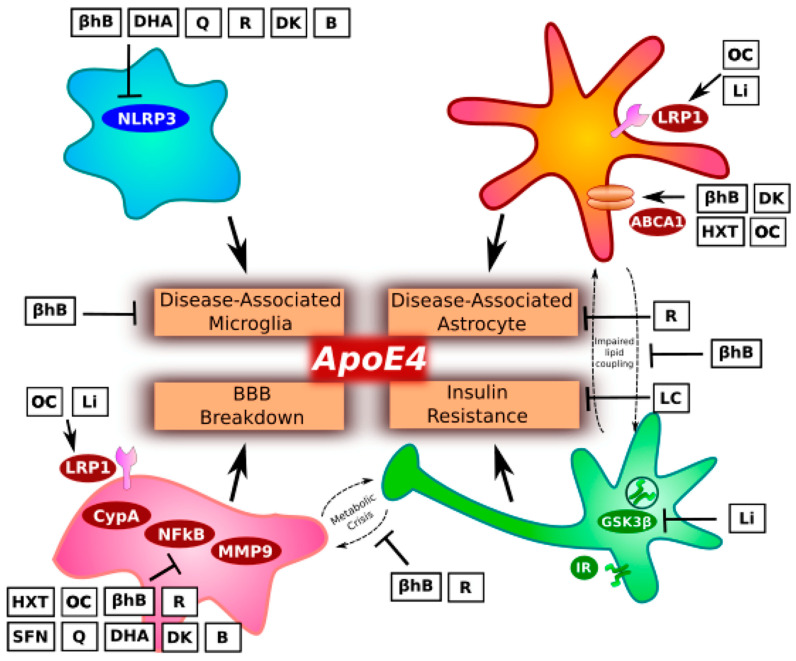
Precision nutrition for *ApoE4* carriers. *ApoE4* can contribute to NLRP3-mediated disease-associated microglia (DAM) formation, disease-associated astrocyte (DAA) formation, activation of the CypA-NFκB-MMP9 pathway in pericytes and loss of blood–brain barrier (BBB) integrity, and insulin resistance. The text described specific nutritional recommendations to target these pathways, including the following options: low-carbohydrate (LC) or ketogenic diets that generate βhB; extra virgin olive oil containing oleocanthal (OC) and hydroxytyrosol (HXT); cruciferous vegetables containing sulphoraphane (SFN); fatty fish containing DHA and DHA supplementation; Quercetin (Q) supplementation, also found in capers and red onions; resveratrol (R) supplementation; vitamins D_3_ and K_2_MK7 (DK) supplementation; vitamin B-complex (B); and low-dose lithium (Li) supplementation.

**Table 1 nutrients-13-01362-t001:** Dietary options and rationales.

Diet Options	Potential Benefits
Low-glycemic index/ Low-carbohydrate diets	Prevent or improve insulin resistance Avoid impairment of Aβ degradation by insulin Avoid GSK3β mediated hyperphosphorylation of tau Avoid reduced cerebral blood flow Reduce AGE formation and ApoE4 glycation, with consequences on cerebral lipid metabolism Low-glycemic index/low-carbohydrate diets are associated with reduced AD prevalence
Ketogenic diet (On the spectrum of low-carbohydrate diets)	Benefits of a low-carbohydrate diet (above) as well as the following: Enhanced lipophagy and protection from lipotoxicity Provides acetyl-CoA to circumvent ‘metabolic crisis’ Increased histone acetylation to promote memory genes Reduced Aβ toxicity Inhibits NLRP3 formation Inhibits CypA-NFκB-MMP9 Improved brain network stability Symptomatic improvements in AD patients
Extra virgin olive oil	Phenolic compounds, oleocanthal and hydroxytyrosol, possess anti-amyloid and anti-tau properties Increases levels of LRP1 Inhibits CypA-NFκB-MMP9 Increase ABCA1
Cruciferous vegetables Capers and red onion	Sulphoraphane decreases NFκB-MMP9 expression Quercetin inhibits NLRP3 formation Quercetin inhibits CypA-NFκB-MMP9
Fatty fish (≥2 times/week)	Inhibits NLRP3 formation Inhibits CypA-NFκB-MMP9 Improves amyloid and tau pathologies of in animal models Greater consumption negatively associated with AD
Limit alcohol	Association data suggest consumption of any alcohol may increase risk in *ApoE4* carriers

**Table 2 nutrients-13-01362-t002:** Supplements and dosing options.

Supplement	Dose
DHA	≥2 g/day
Quercetin	1–2 g/day
Resveratrol	Unclear (2 g/day dose is referenced)
Vitamin D_3_	Up to 5000 IU/day
Vitamin K_2_ MK7	45–180 µg/day
B-vitamin complex	50 mg riboflavin, 0.8 mg folate as methyltetrahydrofolate (5-MTHF), 0.5 mg B_12_ as methylcobalamin, and 20 mg of B_6_ as pyridoxal 5′ phosphate (PLP)
Lithium	Unclear (5 mg is available over the counter)

## Data Availability

Data are contained within the article.

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
