# Peer review of "Precision Nutrition for Alzheimer’s Prevention in ApoE4 Carriers"

_nutrients, 2021, doi:10.3390/nu13041362_

Round 1

Reviewer 1 Report

By reading the manuscript, the first impact I had was that this perspective paper is longer than I would have expected from a “perspective”. The authors should reduce the length of the manuscript that, in the present form, risks to become dispersive.

However, the authors report the integration between basic scientific mechanisms and human observational studies to create a rationale for a useful approach in adopting a lifestyle and nutrition plan aiming at AD’s risk reduction in APOE4 patients. The authors would like to focus on the future of precision nutrition for AD prevention in APOE4 carriers.

This tendency is sometimes lost through the manuscript, as at pag 10, line 400-406. Here, the sentences need to be reformulated in the light of the manuscript’s aim.

Pag 14, line 571-572: please, the resveratrol and “metabolic crisis” concept need to be better explained.

Table 1: this table aims at describing the dietary options and rationale. Specifically, ketogenic diet should exert “benefits above, plus…”. Please, detail. What exactly do the ellipsis in the sentence mean?

When authors discuss the link between APOE4 and mitochondrial dysfunction, they should report the following review, here there is an interesting working model for APOe4 contribution to AD: D’Argenio V et al. J Pers Med. 2020 Apr 19;10(2):26. doi:10.3390/jpm10020026.

Minor points:

Some grammatical and typographical errors should be corrected

Pag3, line 91: replace “minatory” with “minority” I guess

Pag 10, line 385: hB (?)

Pag 10, line 404: please reformulate the sentence “There is mechanistic….”

Author Response

Reviewer 1

Reviewer 1: By reading the manuscript, the first impact I had was that this perspective paper is longer than I would have expected from a “perspective”. The authors should reduce the length of the manuscript that, in the present form, risks to become dispersive. However, the authors report the integration between basic scientific mechanisms and human observational studies to create a rationale for a useful approach in adopting a lifestyle and nutrition plan aiming at AD’s risk reduction in APOE4 patients. The authors would like to focus on the future of precision nutrition for AD prevention in APOE4 carriers.

Thank you for your thoughtful comments . We are pleased you seem to share our opinion that developing a biologically plausible precision nutrition protocol for risk reduction in ApoE4 carriers is a potentially meaningful endeavor.

Regarding the word count, the word limit for this perspective was instructed to be 6,000 -12,000 words, and the current draft is 8000 words. While we agree that it is important to remain concise, we wanted to more thoroughly discuss some of the key recent findings from high-profile publications to provide interested clinicians, researchers, and citizens scientists a somewhat more comprehensive view of where the field stands. Further, we hope that the simplicity of the Tables would serve as a nice contrast and complement to the density of the text but felt that slightly more detail was better given the absence of a similar manuscript in the literature. Nevertheless, we have reviewed the manuscript to remove extraneous sentences and attempt to lower the word count.

Reviewer 1: This tendency is lost at page 10, line 400-406. Here, the sentences need to be reformulated in the light of the manuscript’s aim.

We’ve rewritten what were previously lines 400-406 (now page 11, lines 420-428), along with the opening sentence of that paragraph (page 10, lines 403-405), to clarify their relevance to the topic of prevention and AD risk reduction in ApoE4 carriers.

Reviewer 1: Pag 14, line 571-572: please, the resveratrol and “metabolic crisis” concept need to be better explained.

Thank you for pointing out this issue. We have tried to clarify what we meant by ‘metabolic crisis,’ which we first refer to in section 3 on astrocytes with respect to the 2021 Qi et al findings. However, given the depth and technicality of this piece, reinforcement of the concept is certainly deserved. Thank you for pointing that out. In brief, metabolic crisis refers to the rock and a hard place situation in which glucose and lipid metabolism are both impaired altered by ApoE4 potentially contributing to an energy deficit. BhB (and intermittent fasting) could help protect against lipotoxic damage to mitochondria by promoting lipophagy and BhB also provides an alternative acetyl-CoA precursor for the generation of energy via the Krebs cycle. At the same time, resveratrol could promote PGC1a expression to facilitate further mitochondrial biogenesis and potentially improve deficient glucose metabolism. This is, hopefully, now clarified in the edited text.

Reviewer 1: Table 1: this table aims at describing the dietary options and rationale. Specifically, ketogenic diet should exert “benefits above, plus…”. Please, detail. What exactly do the ellipsis in the sentence mean?

Apologies for the non-descriptive text. “Benefits above plus…” was not clear in Table 1. We’ve changed the bullet point to “Benefits of a low-carbohydrate diet (above) as well as the following:” Thank you for bringing this to our attention.

Reviewer 1: When authors discuss the link between APOE4 and mitochondrial dysfunction, they should report the following review, here there is an interesting working model for APOe4 contribution to AD: D’Argenio V et al. J Pers Med. 2020 Apr 19;10(2):26. doi:10.3390/jpm10020026.

We enjoyed the D’Argenio et al 2020 review. The review was interesting as were the referenced papers (ref [53], Huang 2006, and Harris 2003 referenced therein) that pertained to mitochondrial dysfunction. We’ve added a paragraph on pages 10-11 that references the D’Argenio, Huang, and Harris pieces to address your suggestion that reads, “At minimum, it is biologically plausible that ketogenic diets to could help to prevent AD. BhB can decrease inflammation, oxidative stress, mitochondrial dysfunction, and other basic metabolic pathologies that contribute to metabolic diseases, including AD [83]. As these metabolic pathologies are engaged in a complex positive feedback network, it’s important that any preventative intervention has to potential address modulate metabolism and each node of the network [63, 83]. This may be particularly important in vulnerable ApoE4 carriers. For example, it has been proposed that oxidative stress induces in situ expression of ApoE4 in neurons as a repair response and that ApoE4 is more vulnerable than ApoE3 to proteolytic cleavage, generating a C-terminal ApoE4 fragment that may damage mitochondria [84-86]. Pre-existing mitochondrial dysfunction, inflammation, and insulin resistance could sensitize or exacerbate the positive feedback network. But what if an individual reduced the oxidative stress, inflammation, insulin resistance, and overload metabolic burden through lifestyle? This is, fundamentally, the basis of ApoE4 precision nutrition.” The potential pathogenic role of ApoE fragments is interesting, as was the rest of D’Argenio manuscript. As an aside, we were intrigued to learn NOTCH3 is a candidate for FAD, given its role in Wnt-signaling, which was the topic of another manuscript we wrote on AD in 2019. Thank you for pointing us to this reference.

Reviewer 1: Minor points:

Some grammatical and typographical errors should be corrected

Pag3, line 91: replace “minatory” with “minority” I guess

Pag 10, line 385: hB (?)

Pag 10, line 404: please reformulate the sentence “There is mechanistic….”

Apologies for the grammatical errors and thank you for bringing these to our attention. We have corrected each of the errors listed (now lines 96, 404, and 429, respectively), which includes replacing the statement, “there is mechanistic rationale…” with “At minimum, it is biologically plausible that…”. We have also comprehensively reviewed the manuscript in hopes of removing all other grammatical errors.

Reviewer 2 Report

Nutrigenomics is a buzzing field of research. The authors claim that "Precision nutrition targeting metabolic pathways altered by ApoE4 provides a tool for the potential prevention" of Alzheimer's disease and, after a review of recent research, "speculate a precision-nutrition approach for ApoE4 carriers".

While I share the enthusiasm of the authors for possible lifestyle interventions in the prevention of AD, and I found their review on the physiology of ApoE4 very interesting, I have concerns whether the focus on providing guidelines ApoE4 carriers is warranted based on current evidence, and what the publication of such rather specific guidelines may imply.

  1. There is no consideration of the fact that any recommendations specifically for ApoE4carriers can be useful for individuals only after they know of their genotype. However, medical guidelines and patient advocacy groups strongly discourage predictive testing for ApoE4 in asymptomatic individuals outside of clinical settings where medical genetic counseling is provided. Do the authors advocate direct to consumer genetic testing for ApoE4, based on the sketchy evidence regarding benefits of nutritional interventions in general (and the complete lack of such evidence in ApoE4 carriers in particular) ? At the current state of knowledge, I would consider this unethical. The authors may disagree, but then would need to address in detail the ethics of ApoE4 - testing prominently, weighing pros and cons. Nutrients has recently published a well reasoned review https://pubmed.ncbi.nlm.nih.gov/33065985/ on nutrigenomics, which the authors may not be aware of.
  2. While ApoE4 is a highly interesting focus of research in AD, I did not get the point of focusing on ApoE4, rather than on targetable pathways for AD in general. As stated in the review by Kim et al. (ref. 4 of the present manuscript) "Strong evidence suggests the major mechanism by which apoE influences AD and CAA is via its effects on Aβ metabolism." While ApoE4 leads to earlier amyloid accumulation in the brain, mechanisms of Amlyoid and Tau pathology might be addressed with nutrition irrespective of ApoE4. I could see no argument for devising specific ApoE4 recommendations rather than general recommendations to lower AD / dementia risk (e.g. https://pubmed.ncbi.nlm.nih.gov/33336232/) ?
  3. I tried to find, in the cited references 4 and 5, evidence for the claim that ApoE4 would account for 65% of all AD cases. In very old age, where the vast majority of AD cases occurs, the role of ApoE4 is low. In memory clinic samples of MCI/AD, the population attributable fraction for ApoE4 is very high (69%) only in a small age bracket  65-70 years https://pubmed.ncbi.nlm.nih.gov/32817639/  It seems to me that this exaggerated number serves to increase the relevance of ApoE4 (and an ApoE4-specific preventive diet). 
  4. In would of course be interesting to learn whether the research on ApoE4 has yielded specific pathways not already considered for nutritional interventions for AD. Many of the recommendations have already been clinically tested in AD risk populations (irrespective of ApoE), with largely negative results.
  5. Oftentimes it is not clear if the authors are discussing findings from human research or from animal models and the two sources are mixed within the same paragraphs. This makes it difficult to get a clear idea of the state of the current research in my mind. Would it make sense to divide the evidence coming from human and animal studies? The translational gap between the two should be considered and at least discussed.
  6. For each proposed intervention the Authors propose a rationale supporting their potential for targeting specific disease mechanisms. It would be interesting also to consider the potential side effects and applicability in real life. For example, following a ketogenic diet might be challenging and the systemic consequences should be considered.

In sum, I would recommend that this paper is substantially rewritten, either maintaining the idea of an ApoE4 specific dietary advice, addressing the issues above in depth, or by refocusing on possibly new ideas for nutritional interventions, based on a better understanding of AD mechanisms (also involving ApoE dependent ones).

Author Response

Reviewer 2

Nutrigenomics is a buzzing field of research. The authors claim that "Precision nutrition targeting metabolic pathways altered by ApoE4 provides a tool for the potential prevention" of Alzheimer's disease and, after a review of recent research, "speculate a precision-nutrition approach for ApoE4 carriers".

While I share the enthusiasm of the authors for possible lifestyle interventions in the prevention of AD, and I found their review on the physiology of ApoE4 very interesting, I have concerns whether the focus on providing guidelines ApoE4 carriers is warranted based on current evidence, and what the publication of such rather specific guidelines may imply.

Reviewer 2: There is no consideration of the fact that any recommendations specifically for ApoE4carriers can be useful for individuals only after they know of their genotype. However, medical guidelines and patient advocacy groups strongly discourage predictive testing for ApoE4 in asymptomatic individuals outside of clinical settings where medical genetic counseling is provided. Do the authors advocate direct to consumer genetic testing for ApoE4, based on the sketchy evidence regarding benefits of nutritional interventions in general (and the complete lack of such evidence in ApoE4 carriers in particular) ? At the current state of knowledge, I would consider this unethical. The authors may disagree, but then would need to address in detail the ethics of ApoE4 - testing prominently, weighing pros and cons. Nutrients has recently published a well reasoned review https://pubmed.ncbi.nlm.nih.gov/33065985/ on nutrigenomics, which the authors may not be aware of.

Thank you for taking the time to review our manuscript and provide comprehensive feedback in hopes of improving it. We agree that discussing the ethical implications of genotype testing and disclosure is an important section missing from the manuscript. We have included a three-paragraph discussion, referencing the manuscript you recommended (Mullins, 2020) as well as another RCT that investigated the behavioral impact of disclosing carrier status. The paragraphs are copied below.

“This manuscript is not meant to imply that clinicians should pressure their patients in to getting genetic testing. Given the complete lack of clinical evidence for precision nutrition in the context of AD, such a practice is premature. Furthermore, it is important to acknowledge that ApoE4 is a risk allele likely influenced by gene-gene and gene-environment interactions that might also modify the potential efficacy of the dietary suggestions presented in this text. That said, there may be utility in providing biologically plausible suggestions, as we have done in this manuscript, if for no other reason than direct to consumer genetic testing is becoming more widely available and many patients choose to know their genotype. Potential downsides of direct-to-consumer testing include the potential for misinterpretation of results by patients, as well as a potential negative phycological reaction to the information [176].

The Risk Evaluation and Education for Alzheimer’s Disease (REVEAL) trial was the first randomized controlled to evaluate the impact that disclosure of ApoE4 carrier status had on behavioral change in cognitively normal individuals. The researchers found that individuals who learned they were carriers reported more behavioral changes related to diet, exercise, medications, and vitamins compared to those who learned they were non-carriers [177]. This study demonstrates that the act of disclosing ApoE4 carrier status can positively affect behavior change, which is critical for AD risk reduction success.

While ApoE4 carrier status is a sensitive topic that certainly requires a collaborative discussion between the patient and treating clinician, in cases deemed clinically appropriate by both parties (and where the patient does not already have the information through direct-to-consumer testing), disclosing this information may be a beneficial way to encourage behavioral change. The specifics of this behavior change should be determined only following a thoughtful discussion between the patient and the clinician that considers the individual’s sociocultural and economic circumstances, as well as any other practical limitations in implementing lifestyle change and the personal preferences.”

Reviewer 2: While ApoE4 is a highly interesting focus of research in AD, I did not get the point of focusing on ApoE4, rather than on targetable pathways for AD in general. As stated in the review by Kim et al. (ref. 4 of the present manuscript) "Strong evidence suggests the major mechanism by which apoE influences AD and CAA is via its effects on Aβ metabolism." While ApoE4 leads to earlier amyloid accumulation in the brain, mechanisms of Amlyoid and Tau pathology might be addressed with nutrition irrespective of ApoE4. I could see no argument for devising specific ApoE4 recommendations rather than general recommendations to lower AD / dementia risk (e.g. https://pubmed.ncbi.nlm.nih.gov/33336232/) ?

We chose to write a manuscript specifically on precision nutrition for ApoE4, rather than on risk reduction in the general population, because the allele modifies metabolism and ApoE4 carriers make up a disproportionate amount of the overall burden of AD. Moreover, ApoE4 may confer risk through distinct mechanisms specific to carriers and these can be targeted through biologically plausible precision nutrition interventions. For example, the paired Montagne 2020 and Bell 2012 papers discussed in the pericyte section shows that ApoE4 can induce blood-brain barrier breakdown via the CypA-MMP9 pathway that is independent of amyloid and tau pathology (thus going beyond the amyloid model) and that predicts cognitive decline in carriers, but not non-carriers. Other data suggests that the interventions we discussion in section six of the text can target that specific pathway, possible reducing risk in ApoE4 carriers, specifically. In general, the goal of the manuscript was to discuss the new data investigating mechanisms – often independent of amyloid – by which ApoE4 may confer risk and then to discuss low-risk dietary options to target these pathways.

Reviewer 2: I tried to find, in the cited references 4 and 5, evidence for the claim that ApoE4 would account for 65% of all AD cases. In very old age, where the vast majority of AD cases occurs, the role of ApoE4 is low. In memory clinic samples of MCI/AD, the population attributable fraction for ApoE4 is very high (69%) only in a small age bracket  65-70 years https://pubmed.ncbi.nlm.nih.gov/32817639/  It seems to me that this exaggerated number serves to increase the relevance of ApoE4 (and an ApoE4-specific preventive diet). 

We apologize for the misrepresentation, it was not our intention to present an exaggerated statistic and we agree that the proportion of AD cases associated with ApoE4 will depend on the population begin studied, including age and ethnicity. Therefore, we’ve changed the statement (page 2, line 63) to reflect an estimated range, 40 – 65%, reported by the Alzheimer’s Association and referenced that group. If you prefer, we can omit that sentence entirely.

Reviewer 2: In would of course be interesting to learn whether the research on ApoE4 has yielded specific pathways not already considered for nutritional interventions for AD. Many of the recommendations have already been clinically tested in AD risk populations (irrespective of ApoE), with largely negative results.

For the most part, the nutritional interventions we recommend haven’t been tested for AD risk reduction in controlled trials. The existing data are primarily epidemiological associations. As discussed in greater depth in the third to last paragraph in the introduction, there are several controlled trials (only one that performed an ApoE4 subgroup analysis), but all are limited by being multidomain interventions and initiating protocols in older individuals >65 years in whom the disease process would have likely already commenced. It is unlikely that any true prevention dietary trials will be conducted given the logistics of conducting such a study. Therefore, we aimed to offer a next best option for ApoE4 carriers, a set of biologically plausible dietary suggestions that could address genotype specific defects reported in the recent literature.

Reviewer 2: Oftentimes it is not clear if the authors are discussing findings from human research or from animal models and the two sources are mixed within the same paragraphs. This makes it difficult to get a clear idea of the state of the current research in my mind. Would it make sense to divide the evidence coming from human and animal studies? The translational gap between the two should be considered and at least discussed.

Thank you for bringing this point to our attention. We have  gone through the microglia, astrocytes, pericytes, and insulin resistance sections of the manuscript and commented on the model systems from which the data were drawn. We have also included two statements, in the introduction(page 3, lines 107-110) and at the beginning of the recommendations section (page 8, lines 353-356) to emphasize that the research is predominantly animal models and the complexity of translating these evidence for human purposes. We agree it is important to comment directly on the translational gap. It is an unfortunate limitation of any manuscript based on a bottom-up first principles approach but we agree that it is important to acknowledge. We also went back and may sure to highlight when we were discussing post-mortem or in vivo human data.

Reviewer 2: For each proposed intervention the Authors propose a rationale supporting their potential for targeting specific disease mechanisms. It would be interesting also to consider the potential side effects and applicability in real life. For example, following a ketogenic diet might be challenging and the systemic consequences should be considered.

Thank you for bringing this to our attention. We have included additional text (page 18, lines 731-735) to encourage patients and the general population to consult with medical professionals before taking on any of our suggestions— “The specifics of this behavior change should be determined only following a thoughtful discussion between the patient and the clinician that considers the individual’s sociocultural and economic circumstances, as well as any other practical limitations in implementing lifestyle change and the personal preferences.”

We very much appreciate the time and effort that you evidently put into providing us with feedback. The adapted manuscript represents our best effort at balancing your comments against those of the other reviewers, who valued the precision nutrition for ApoE4 concept, albeit speculative. Each of your comments were of great value, and we particularly appreciate your suggestion of an ethics statement that clarifies we are not trying to prescribe an approach, nor are we pressuring clinicians or individuals into getting genotype testing. Instead, we only aim to provide an evidence and mechanism-informed perspective. In our research and clinical experience, there are enough people looking for such a resource that we feel this manuscript will positively contribute to the field.

Reviewer 3 Report

The manuscript is in general well written and clear. The aim of the authors is to discuss data on diet effects in reducing risk for AD in ApoE4 carriers,. They focused on the multiple influences of ApoE on microglia, astrocytes, pericytes  vascular, insulin and lipid metabolism. For each of these mechanims they indicate the potential role of a correct diet highlighting the potential effect of the Mediterranean life style. 

The concept is not new, the idea to differentiate among various paths is interesting. 

NO major concerns for the article 

Author Response

Thank you for your generous comments. We are glad you appreciated the manuscript and hope that it will provide individuals and clinicians some insight, options, or food for thought at minimum.  

Round 2

Reviewer 2 Report

The authors have revised the manuscript reasonably well.      Two minor remaining issues:   1. The new ethics section is appreciated. However, the first sentence of this section is rather bizarre „This manuscript is not meant to suggestimply that clinicians should pressure or insist that their patients in to gettingundergo genetic testing , as clinicians NEVER should, or will, pressure or insist on such testing. So there is no point here to emphasize the obvious. How about something like: „We do not advocate broad genetic testing for ApoE4. Given the lack of …“   2. Could the authors perhaps include a sentence on the overlap of their ApoE4 related suggestions with other „dementia-prevention“ dietary recommendations? I think that this overlap is considerable. This would also allow to deemphasize ApoE4 testing as a prerequisite of applying the recommendations.    I found some typos in the revised paragraphs, please check.

Author Response

Thank you for these comments, both of which we were happy to address. We've changed the first line (688) of the 'Ethical Considerations' Section to "This manuscript is not meant to advocate broad genetic testing for ApoE4."

We've also added the following two sentences at line 705-710, "That said, patients do not require knowledge of their ApoE status to implement the dietary or supplement options provided in Tables 1 and 2. It is also likely that several of the suggestions, such as consuming fatty fish, are likely to benefit the cognitive longevity of most persons. Thus, a conservative patient-clinician team could implement dietary options without genetic testing."

Apologies for the typos. Those are not present in out April 6 copy of the manuscript and may have arisen out of a copy-paste error with tracked changes. In any case, we do not see them in the word or PDF versions of the manuscript at this point.

Thank you again for the time and effort you put into helping us improve this manuscript.